# Metabolic Alterations in Pancreatic Cancer Detected by In Vivo ^1^H-MR Spectroscopy: Correlation with Normal Pancreas, PET Metabolic Activity, Clinical Stages, and Survival Outcome

**DOI:** 10.3390/diagnostics11091541

**Published:** 2021-08-25

**Authors:** Chih-Kai Chang, Tiffany Ting-Fang Shih, Yu-Wen Tien, Ming-Chu Chang, Yu-Ting Chang, Shih-Hung Yang, Mei-Fang Cheng, Bang-Bin Chen

**Affiliations:** 1Department of Medical Imaging, National Taiwan University Hospital, Taipei 100, Taiwan; kent90409@gmail.com (C.-K.C.); ttfshih@ntu.edu.tw (T.T.-F.S.); 2Department of Radiology, College of Medicine, National Taiwan University, Taipei 100, Taiwan; 3Department of Surgery, National Taiwan University Hospital, Taipei 100, Taiwan; ywtien5106@ntu.edu.tw; 4Department of Internal Medicine, National Taiwan University Hospital, Taipei 100, Taiwan; mingchuchang@ntu.edu.tw (M.-C.C.); yutingchang@ntu.edu.tw (Y.-T.C.); 5Department of Oncology, National Taiwan University Hospital, Taipei 100, Taiwan; shyang0821@ntuh.gov.tw; 6Department of Nuclear Medicine, National Taiwan University Hospital, Taipei 100, Taiwan; meifang@ntuh.gov.tw

**Keywords:** MR spectroscopy, positron emission tomography, pancreatic cancer, survival

## Abstract

Objective: To compare the metabolites of in vivo 1H- MRS in pancreatic cancer with normal pancreas, and correlate these metabolites with Positron Emission Tomography (PET) metabolic activity, clinical stages, and survival outcomes. Methods: The prospective study included 58 patients (mean age 62.7 ± 12.1 years, range 34–81 years; 36 men, 22 women) with pathological proof of pancreatic adenocarcinoma, and all of them received 18F-fluorodeoxyglucose (FDG) PET/MRI before treatment. The single-voxel MRS with a point-resolved selective spectroscopy sequence was used to measure metabolites (creatine, Glx (glutamine and glutamate), N-acetylaspartate (NAA), and lipid) of pancreatic cancer and adjacent normal parenchyma, respectively. FDG-PET parameters included SUVmax, metabolic tumor volume (MTV), and total lesion glycolysis (TLG). Non-parametric tests were used to evaluate the differences of MRS metabolites between pancreatic cancer and those in normal pancreas, and their correlation with PET parameters and clinical stages. The correlation with progression-free survival (PFS) and overall survival (OS) was measured using the Kaplan–Meier and Cox proportional hazard models. Results: When compared with normal pancreas, the Glx, NAA, and lipid levels were significantly decreased in pancreatic cancer (all *p* < 0.05). Creatine, Glx, and lipid levels were all inversely correlated with both MTV (rho = −0.405~−0.454) and TLG (rho = −0.331~−0.441). For correlation with clinical stages, lower lipid levels were found in patients with T4 (vs. <T4, *p* = 0.038) and lower creatine levels were found in N1 (vs. N0, *p* = 0.019). Regarding survival outcomes, high TNM stage, low creatine, low Glx, and low lipid levels were associated with both poor PFS and OS (all *p* < 0.05). Additionally, creatine remained an independent factor for PFS and OS after adjusting for age, sex, tumor size, stages, and other metabolites levels. Conclusions: Decreased MRS metabolites in pancreatic cancer were associated with poor survival outcome, and may be used as prognostic image biomarkers for these patients.

## 1. Introduction

Pancreatic cancer accounts for the fourth and sixth leading causes of cancer-related mortality in the US and China, respectively [1,2]. Diagnostic imaging is essential for early diagnosis and treatment planning in these patients. Conventional imaging modalities, such as CT or MRI, provide valuable information on adjacent organ involvement, but are limited in revealing biologic information. Molecular imaging techniques, such as MR spectroscopy (MRS), provide a wide range of metabolic and functional information, which helps differentiate benign tumors from malignancy, evaluate aggressiveness, and predict prognoses in cancer patients [3].

MRS assesses the resonance frequencies that result from electronic clouds around atoms and identifies certain metabolites by signals that result from differences in resonant frequencies [3]. The common types of metabolites include choline-containing compounds (which are used to make cell membranes); lipid metabolism-related compounds (methylene (–CH_2_) and methyl (CH_3_)); creatine (Cr), glutamine (Gln), and glutamate (Glu) (which are involved in energy metabolism); and N-acetyl-aspartate (NAA). For example, NAA and citrate can be detected in the normal brain and normal prostate, respectively, and their levels are decreased when such tissue becomes a tumor. Total choline has been used to diagnose and monitor treatment responses in breast, brain, and prostate cancers. High levels of lipid in high-grade gliomas are related to apoptosis, necrosis, or lipid droplet formation [4].

Recent studies have proven that metabolic alterations can promote pancreatic tumorigenesis and metastasis through epigenetic regulation [5]. Pancreatic cancer cells can rewire many metabolic pathways to facilitate their growth, modify interactions with the extracellular matrix within the tumor microenvironment, and even affect host antitumor immunity via cachexia [6]. Furthermore, cancer cell metabolism is closely associated with chemoresistance, radioresistance, and immunosuppression. Some studies also stratified pancreatic cancer into different metabolic subgroups (quiescent, glycolytic, cholesterogenic, and mixed) to predict prognoses and responses to therapy [7,8]. Therefore, the metabolic characteristics of pancreatic cancer may offer new insights and opportunities for personalized treatment [9].

There are few studies that used MRS metabolites to predict survival outcomes in pancreatic cancer patients [10,11]. We hypothesize that in vivo ^1^H-MRS metabolites can detect alterations in metabolism in pancreatic cancer and that these metabolites are associated with prognosis in these patients. The purpose of this study is to compare the metabolites of in vivo ^1^H-MRS in pancreatic cancer patients with adjacent normal pancreatic parenchyma, and correlate these metabolites with PET parameters, clinical stages, and survival outcomes.

## 2. Materials and Methods

Approval from our institutional review board was obtained for this prospective study. Informed consent was obtained from all patients.

### 2.1. Patient Population

From August 2014 to July 2016, 106 consecutive patients were admitted for examination by PET/MRI before treatment. The inclusion criteria were suspicious pancreatic cancer on sonography or CT and no local or systemic therapies. The exclusion criteria were pregnancy and contraindications for MRI. All the patients’ pathologies were proven by surgery, endoscopic ultrasound, or CT-guided biopsy. Tumor size, histological grade, and lymph node metastasis data were recorded. The TNM stages (American Joint Committee on Cancer, 7th edition) of the study patients were determined by a multidisciplinary team for pancreatic cancer at our hospital. Finally, 58 patients (mean age 62.7 ± 12.1 years, range 34–81 years; 36 men, 22 women) with pancreatic ductal adenocarcinoma comprised our study population after excluding 48 patients (chronic pancreatitis, *n* = 15; other malignancy, *n* = 28; poor MRS quality, *n* = 4; and no normal pancreatic parenchyma identified, *n* = 1) (Figure 1). Patient characteristics are presented in Table 1.

The association of choline in pancreatic cancer with histological grades, clinical stages, and survival outcomes had been reported previously [10,11], but no association of choline with prognosis was found. In this study, we focus primarily on the associations between other MRS metabolites and clinical outcomes and pathologic findings. We also compared the levels of these MRS metabolites between pancreatic cancer and normal pancreas tissue, which have not been reported previously.

### 2.2. Image Acquisition and Analysis

All patients received examination in a 3T PET/MR (Biograph mMR; Siemens Healthcare, Erlangen, Germany). ^18^F-fluorodeoxyglucose (FDG)-PET was performed from the head to the mid-thighs in 5 bed positions (acquisition time, 4 min/position) with the patient in a supine arm-down position. Images were reconstructed using an ordered-subsets expectation maximization iterative algorithm (2 iterations, 21 subsets) with a 5-mm post-reconstruction Gaussian filter and a 172 × 172 image matrix. Attenuation correction of the PET data was performed using a 4-tissue-class (air, lung, fat, and soft tissue) segmented attenuation map acquired using a 2-point Dixon MRI sequence.

Single-voxel MRS data were acquired by replacing the volume of interest in pancreatic cancer and normal pancreatic parenchyma, with careful avoidance of inclusion of adjacent vessels. The voxel sizes used for pancreatic cancer and normal pancreatic parenchyma in each patient were different according to the size of the tumor and normal pancreatic parenchyma. A point-resolved selective spectroscopy sequence (1000/30 ms TR/TE; 90° flip angle) was used during free-breathing. Automated optimization of the transmitter pulse power, localized shimming, gradient tuning, and water suppression was used. The spectral bandwidth was 1200 Hz, and 200 signals were averaged. Fat-saturation bands were not used. The acquisition time for MRS was 200 s. 

One radiologist (B.B.C., with 12 years of experience in abdominal imaging) measured the MRS data. The analysis was calculated using syngo.MR Spectroscopy software (Siemens Healthineers, Erlangen, Germany). This software automatically postprocessed the data via the following steps: (1) identification of the prominent metabolites by cross-correlation to a database, (2) determination of the B_0_ shift and starting values for the fit parameters, (3) residual water removal by an iterative low-pass filter around the frequency at 4.7 ppm, (4) fit of the spectroscopic data based on a basis set of metabolic model signals, and (5) truncating and remodeling of the first data points to handle baseline artifacts. The volumes of interest were 3.6 ± 1.3 cm^3^ (range, 3.4–12.5 cm^3^) for tumor and 3.9 ± 1 cm^3^ (range, 1.9–7.7 cm^3^) for normal pancreatic parenchyma.

The following metabolites were in the fit (Figure 2A,B):1.Creatine (Cr) with separately fitted CH_2_ (Cr_2, 3.09 ppm) and CH_3_ (Cr_1, 3.03 ppm) groups;2.N-Acetylaspartate (NAA, 2.02 ppm);3.Glutamate (Glu) and Glutamine (Gln) (2.05–2.50 ppm): Glx as a combined Glu/Gln after separately fitted with each amplitude;4.Two lipid/macromolecule lines with separately fitted CH_3_ (Lipid_1, 0.9 ppm) and CH_2_ (Lipid_2, 1.5 ppm) groups.

PET-related parameters were measured by the same radiologist: SUV_max_, which reflects the maximal standard uptake value (adjusted for body weight); metabolic tumor volume (MTV), expressed as the tumor volume with FDG uptake, which was segmented using a fixed-percentage threshold method at 50% of the SUV_max_; and total lesion glycolysis (TLG), representing the product of MTV and the average SUVs of the included voxels. The fixed-threshold MTV and TLG were automatically derived from tumor delineations by the software (Syngovia tool, Siemens Healthineers, Erlangen, Germany).

### 2.3. Statistical Analysis

Summary statistics are presented as the mean (±standard deviation) for continuous variables or frequency and percentage for categorical variables. The paired Wilcoxon signed-rank test was used to compare MRS metabolite levels between pancreatic cancer and normal pancreatic parenchyma. The Mann–Whitney test was used to compare differences in the MRS metabolites relative to tumor grade and TNM stages. Spearman correlation (rho) analysis was used to determine the correlation between MRS metabolites and PET parameters. Progression-free survival (PFS) was defined as the time from MRI to the time of tumor progression. Overall survival (OS) was measured from the date of the MRI to the date of death or study completion (29 February 2020). For the survival analysis, each variable was dichotomized as either high or low, based on the median value of the variable. The optimal cutoff for the predictor was estimated by using the maxstat [12] function in R statistical software (R, version 4.0.2; R Foundation for Statistical Computing, Vienna, Austria). The Kaplan–Meier method was used to plot survival curves and the two-sided log-rank test was used to assess differences in OS and PFS between the patient groups. Multivariable analysis was investigated using the step-wise backward Cox proportional hazard model with several variables, including age, sex, tumor size, TNM stage, and MRS metabolites. Data were analyzed using SPSS software (SPSS for Windows 22; SPSS, Chicago, IL, USA) and R software. Statistical significance was recognized when *p* < 0.05.

## 3. Results

### 3.1. Clinical Treatment and Follow-Up

Of all 58 patients, 16 (28%) received curative surgery (Whipple operation, *n* = 14; distal pancreatectomy, *n* = 2) and 14 received adjuvant chemotherapy. In the 42 patients who did not receive curative surgery, 34 received chemotherapy and 8 received conservative treatment (Table 1). Of 58 patients, 50 (86%) died during this study; the remaining 8 patients had no tumor recurrence. The 1-year, 3-year, and 5-year OS rates were 39.7%, 12.1%, and 1.7%, respectively. The median PFS was 3.0 months (95% confidence interval (CI) = 2.1–3.8 months), median OS was 9.8 months (95% CI = 7.2–12.3 months), and median follow-up was 53.9 months (95% CI = 53.4–54.1 months).

### 3.2. Comparison of MRS Metabolites between Pancreatic Cancer and Normal Pancreatic Parenchyma

When compared with normal pancreatic parenchyma, pancreatic cancer had significantly decreased Glx (*p* = 0.009), NAA (*p* = 0.001), Lipid_2 (*p* = 0.042), and Lipid_1 (*p* < 0.001) levels. Additionally, the Cr_1 level was decreased in pancreatic cancer (*p* = 0.051) (Table 2).

### 3.3. Correlation of MRS Metabolites in Pancreatic Cancer with Pathologic Grade and Clinical TNM Stage

No differences were found in the MRS metabolite levels between well-moderately and poorly differentiated tumors (Table 3).

When correlated with clinical stages, lower lipid_1 level was found in T4 patients (vs. <T4, *p* = 0.038) and lower Cr_2 level was found in N1 patients (vs. N0, *p* = 0.019) (Table 3).

### 3.4. Correlation of MRS Metabolites in Pancreatic Cancer with PET Parameters

The PET related parameters of pancreatic cancer were 5.9 ± 3.3 g/mL for SUV_max_, 11.3 ± 13.6 (cm^3^) for MTV, 62.6 ± 111 (g) for TLG.

SUV_max_ showed weak inverse correlations with NAA (rho = −0.26, *p* = 0.045) (Figure 3).

MTV showed inverse correlations with Cr_1 (rho = −0.441, *p* < 0.001), Glx (rho = −0.348, *p* = 0.004), lipid_2 (rho = −0.340, *p* = 0.005), and lipid_1 (rho = −0.331, *p* = 0.006).

TLG showed moderate inverse correlations with Cr_1 (rho = −0.418, *p* = 0.001), Glx (rho = −0.405, *p* = 0.01), lipid_2 (rho = −0.454, *p* = < 0.001), and lipid_1 (rho = −0.415, *p* = 0.001).

### 3.5. Relationships between Clinical Parameters and MRS Metabolites with Survival Outcomes

Univariate analyses revealed that high TNM stage was prognostic factors for poor PFS (*p* = 0.014; hazard ratio (HR) = 2.096, 95% CI = 1.164–3.774) and OS (*p* = 0.037; HR = 1.858, 95% CI = 1.038–3.326). 

For MRS metabolites, high Cr_2 (*p* = 0.022; HR = 0.32, 95% CI = 0.115–0.895), Cr_1 (*p* = 0.01; HR = 0.466, 95% CI = 0.256–0.846), Glx (*p* = 0.011; HR = 0.472, 95% CI = 0.261–0.854), and lipid_1 (*p* = 0.012; HR = 0.452, 95% CI = 0.24–0.851) were prognostic factors for longer PFS (Figure 4A). Additionally, high Cr_2 (*p* = 0.039; HR = 0.039, 95% CI = 0.096–1.001), Cr_1 (*p* < 0.0001; HR = 0.313, 95% CI = 0.17–0.576), Glx (*p* = 0.00048; HR = 0.449, 95% CI = 0.254–0.794), lipid_2 (*p* = 0.016; HR = 0.489, 95% CI = 0.27–0.887), and lipid_1 (*p* = 0.014; HR = 0.478, 95% CI = 0.261–0.873) were prognostic factors for longer OS (Figure 4B). There was a trend of low NAA level with poor OS (*p* = 0.052).

In multivariable analysis, both TNM stage (*p* = 0.023) and Cr_2 (*p* = 0.025) remained independent predictors of PFS; Cr_1 (*p* = < 0.0001) remained independent predictor of OS (Table 4).

### 3.6. Subgroup Analysis in Patients with and without Curative Surgery

As curative surgery was a significant predictor for both PFS (*p* = 0.021) and OS (*p* = 0.011), we performed a subgroup analysis of patients with and without curative surgical treatment.

In patients receiving curative surgery (*n* = 16, 28%), low Cr_1 level was significant predictor for OS (*p* = 0.0201; HR = 0.219, 95% CI = 0.06–0.789).

In patients without curative surgery (*n* = 42, 72%), low Cr_1 (*p* = 0.0176; HR = 0.4377, 95% CI = 0.221–0.866) and low lipid_1 (*p* = 0.0467; HR = 0.4846, 95% CI = 0.237–0.989) were both significant factors for poor OS.

## 4. Discussion

Our results showed that the levels of several MRS metabolites in pancreatic cancer were significantly decreased compared to those of normal pancreas, and these metabolite levels were inversely correlated with PET metabolic activity. For survival outcome, low Cr, Glx, and lipid levels suggested poor PFS and OS in these patients. Additionally, Cr remained an independent predictor of PFS and OS in the multivariable analysis after adjusting for age, gender, tumor size, TNM stage, and other MRS metabolite levels. Even in the subgroup analysis according to curative surgery, Cr remained a significant prognostic factor for OS. Thus, these metabolites, especially Cr, may be used to identify patients who need more intensive treatment and close follow-up after treatment.

Cr (3.03 ppm) is a marker of energy metabolism as it is utilized as an energy reservoir in cells with high energy demand after phosphorylation. Phosphocreatine can donate phosphoryl groups for ATP synthesis when energy supplies are low [13]. The total Cr (sum of combined Cr and phosphocreatine) concentration is decreased in most tumors, when compared to their normal tissues of origin [14]. In our study, Cr_1 was decreased in pancreatic cancer (*p* = 0.052) and low Cr_2 was found in N1 tumors (*p* = 0.019), suggesting a more aggressive behavior in tumors with low Cr levels. Papalazarou et al. [15] also found that the Cr–phosphagen ATP-recycling system has a role in the invasive migration, chemotaxis, and liver metastasis of pancreatic cancer cells. Consequently, low Cr in pancreatic cancer was also found to be associated with poor PFS and OS in our study.

NAA (2.02 ppm) is a marker of neuronal viability and is the second-most abundant amino acid compound in the human central nervous system (CNS) after glutamate. It is synthesized in mitochondria by aspartate N-acetyltransferase from acetyl–coenzyme A and aspartate. Outside the CNS, NAA also plays a crucial role in the development of several pathological conditions. In the pancreas, NAA serves as a modulator of pancreatic insulin secretion, which is related to adipocyte and whole-body energy homeostasis [16]. The NAA pathway also has a role in promoting tumor growth and may be a valuable target for anticancer therapy [17]. In several tumors (lung, breast, and ovarian cancers), the amount of NAA was increased and inversely correlated with patients’ survival [18]; however, in our study, NAA levels were decreased in pancreatic cancer and seemed to be associated with poor OS (*p* = 0.052). Therefore, many questions remain open when it comes to the functional role of NAA in pancreatic cancer, and investigating the underlying mechanism may reveal novel roles for NAA.

Glx (2.05–2.50 ppm) is a composite peak that incorporates Glu and Gln. The separation between these two metabolites is unreliable, although the sum (Glx) can be quantified with high accuracy. Glu and Gln are in a dynamic balance. While Gln usually fuels the citric acid cycle for oxidative phosphorylation and macromolecular biosynthesis, pancreatic cancer cells rely on non-canonical utilization of Gln to maintain their reactive oxygen species homeostasis [19]. Additionally, Gln biosynthesis by Glu ammonia ligase is also activated in pancreatic cancer cells, adding another anabolic pathway that is critical for tumor growth [20]. In high-grade pancreatic cancer, Glx is markedly consumed to support nucleotide biosynthesis and adenosine triphosphate (ATP) production. Roux et al. also found that a decrease in endogenous Glu is associated with pancreatic cancer progression [21]. Additionally, inhibition of the downstream components of Gln metabolism leads to a decrease in tumor growth [22]. Targeting Gln metabolism could disrupt cancer cell metabolic reprogramming in multiple ways and may represent an effective therapeutic approach for pancreatic cancer [23]. In the future, MRS could be used to monitor treatment response in patients receiving novel therapeutic agents that target tumor metabolism.

The lipid (0.8–1.5 ppm) signal was mainly comprised of narrow signals at 1.3 ppm, which arose from methylene groups (–CH2–CH2–CH2–), and at 0.9 ppm, which arose from methyl groups (CH3–CH2–). These methylene and methyl signals originate from the fatty acyl chains of triacylglycerides that form mobile lipid droplets in the cytoplasm or in the extracellular space [24]. Lipid levels are usually high in a healthy pancreas, but may drop significantly in the presence of inflammation, precancerous metaplasia, pre-invasive pancreatic intraepithelial lesions, or invasive pancreatic cancer [25,26]. In our study, lower lipid levels were observed in high T stage (>T3) patients, and were correlated with poor PFS and OS. A transcriptomics and metabolomics study revealed that the levels of lipase and a panel of fatty acids are significantly decreased in pancreatic tumors [27]. Additionally, two saturated fatty acids, palmitate, and stearate, can inhibit the proliferation of pancreatic cancer cells [27]. Thus, fatty acid metabolism may play an essential role in pancreatic cancer proliferation [9].

FDG-PET can provide information of both tumor metabolism and metastatic extent, and has been listed as an optional imaging modality of preoperative evaluation in current pancreatic cancer guidelines. We found that Glx, Cr, and lipid levels were all inversely correlated with both MTV and TLG. Additionally, NAA showed weak inverse correlations with SUV_max_. The stronger correlation of MRS metabolites with both MTV and TLG rather than SUV_max_ may be attributable by the fact that MRS, MTV, and TLG were all volumetric parameters, whereas SUV_max_ only represented the single hottest pixel value of the tumor. 

Pancreatic cancer could be stratified into different metabolic subgroups (quiescent, glycolytic, cholesterogenic, and mixed). A glycolytic subtype indicates poor survival, whereas a cholesterogenic subtype correlates with better outcomes [7,8]. It is possible to use PET/MRS for similar metabolic subtyping. We assume that glycolytic subtype shows high FDG uptake on PET, whereas cholesterogenic subtype displays high lipid metabolism-related compounds (Cr, Glx, and lipid) on MRS. Consequently, tumors with increased FDG activity had been shown to correlate with poor survival [28], while tumors with high Cr, Glx, and lipid levels showed better outcomes in this study. Thus, the integration of PET/MRS information holds great promise for developing efficacious treatments by optimally targeting metabolic pathways in different subtypes of pancreatic cancer. 

The main disadvantages of MRS are long acquisition times for data collection, lack of adequate signal-to-noise ratio, sometimes improper water/lipid signal suppression, and limited spatial coverage [29]. Additionally, the separation of metabolite signals in the spectrum is affected by field homogeneity, so high-field MRI is necessary to achieve better image quality. A standardized protocol with fine-tuned, optimized signal-to-noise ratios and repeated measurement is necessary to achieve reliable results. Despite these drawbacks, recent research strongly suggests a promising future for MRS in the management of oncologic patients [30]. Based on our study, in addition to clinical stages, pancreatic cancer patients with low Cr levels may need a more aggressive treatment strategy to improve OS rates. As MRS technology continues to advance, and knowledge of tumor biology increases, the use of MRS will help us to gain greater insight into tumor characteristics and predict patient prognoses in the precision medicine era.

In vivo MRS is a complex radiological assay due to inconsistencies between measurements resulting from low signal-to-noise and large linewidths. The linewidth directly affects both the resolution and signal-to-noise of each spectral resonance. Signal-to-noise could be worse in the pancreatic cancer voxels due to heterogeneity in the tissue, which leads to inhomogeneity of the magnetic field. The abdomen’s air, water, and tissue interfaces would make generating a homogenous magnetic field challenging. A future study is needed to correlate metabolite concentration with immunohistochemistry staining of proteins associated with a particular metabolite, such as NAA with NAT8L. A previous study has shown excellent agreement between histological measures of fat content from the resected pancreas and MRI quantification of the fat fraction [31]. Ex vivo MRS of biopsy or resected pancreatic tissue is necessary to illustrate the same reduction in Cr, NAA, or lipids observed by in vivo MRS.

Recent experts’ recommendations on advanced MRS techniques have been reported [32,33], including implementing advanced localization sequences, incorporating simulated metabolite basis sets in spectral analyses, and optimizing algorithms for automated shimming. Standardized protocols following these recommendations are necessary for future multi-institutional longitudinal research. We expect these advanced MRS techniques to substantially increase the data quality and contribute to optimal treatment selection in the clinical setting.

Our study has several limitations. First, we did not analyze all MRS metabolites, such as myoinositol or glutathione, as they were only detected in less than half of our patients. Second, the lactate methyl resonance peak also overlapped with the lipid methylene resonance, which was not analyzed in our study. In 3T MRI, the detection of tumor lactate concentrations using conventional MRS techniques is still a challenging task, and spectral editing is required to separate the methylene signals from the lactate signal [34]. Third, we did not perform a test/retest study or the same analysis on a phantom to illustrate the accuracy of the measurement. A previous study has shown that test–retest measurements of pancreatic fat content calculated using MRS were repeatable [35]. We compared MRS in pancreatic cancer and adjacent normal pancreas in the same scan, which could minimize measurement inconsistencies and the effect of inhomogeneity of the magnetic field on MRS results. Fourth, we did not analyze the correlation between MRS metabolite values and voxel size, and small voxel size may influence the measurement results. Finally, the survival outcomes of cancer patients may be affected by different treatments. Further study with larger patient populations is needed to identify the prognostic value of these metabolites in different treatment subgroups.

In conclusion, pancreatic cancer had decreased levels of metabolites when compared to normal pancreatic parenchyma, suggesting a higher energy demand and increased metabolite consumption in the cancer cells. These metabolites, especially Cr, may be used as prognostic imaging biomarkers to select the most appropriate treatment strategy in these patients.

## Figures and Tables

**Figure 1 diagnostics-11-01541-f001:**
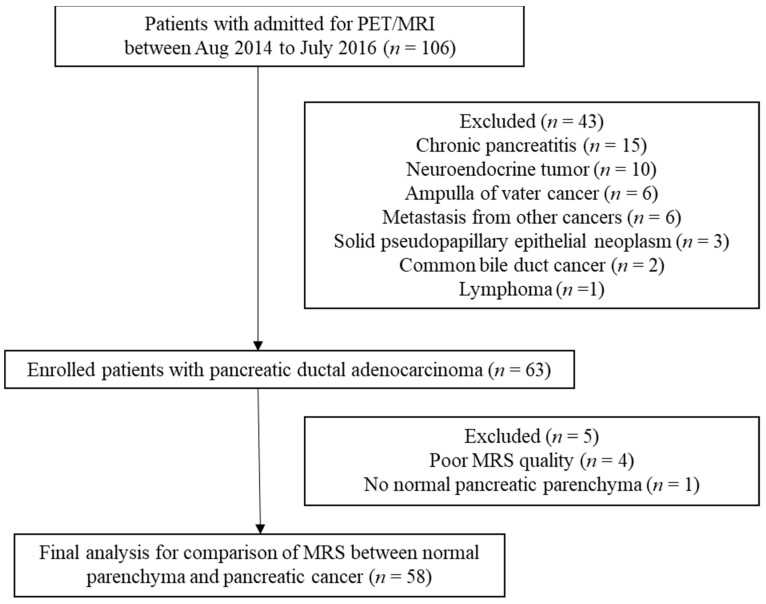
Flowchart of study design.

**Figure 2 diagnostics-11-01541-f002:**
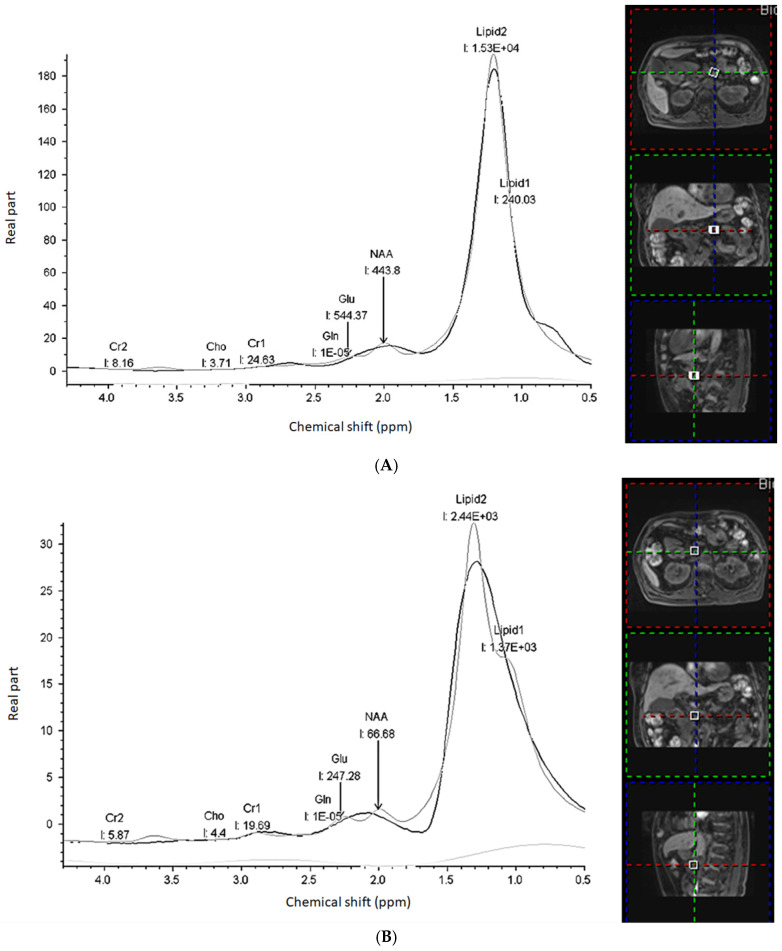
MRS images for a 71-year-old woman with pancreatic head cancer (T3N1M1). The progression-free survival and overall survival are 4.2 and 10.5 months, respectively. When compared to normal pancreas (**A**), the pancreatic cancer (**B**) has decreased Lipid_1, Lipid_2, NAA, Cr_1, Cr_2, and Glx.

**Figure 3 diagnostics-11-01541-f003:**
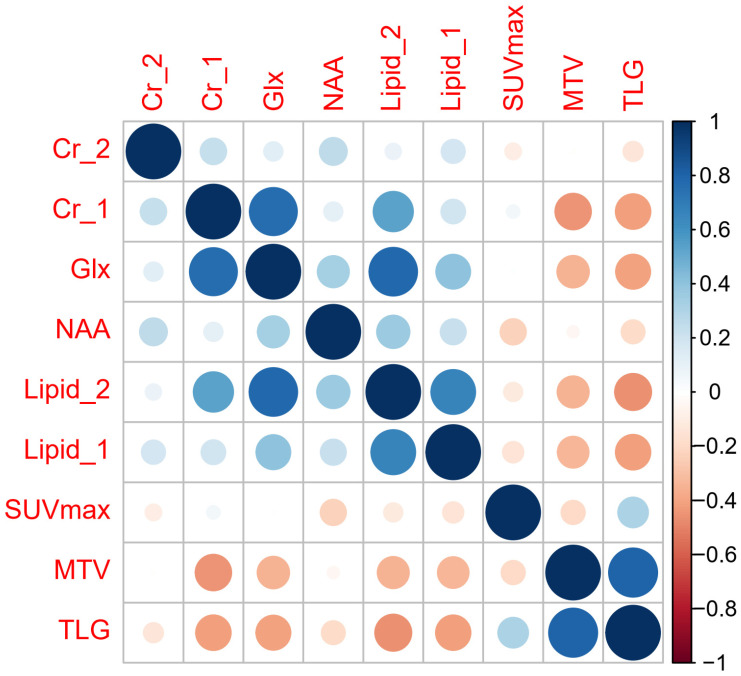
Correlation matrix of Spearman’s rank correlation coefficients. Size corresponds to strength of correlation.

**Figure 4 diagnostics-11-01541-f004:**
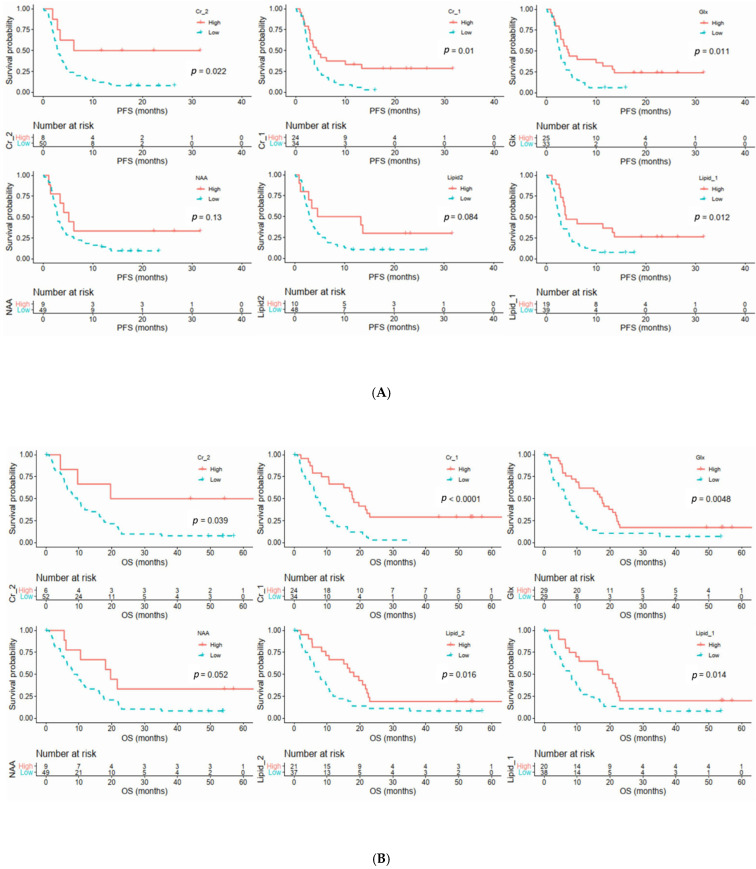
Kaplan–Meier curves of MRS metabolites of PFS (**A**) and OS (**B**).

**Table 1 diagnostics-11-01541-t001:** Demographic and clinical characteristics of the 58 participants.

Parameter	Variable
Age (years) *	62.7 ± 12.1 (34~81)
Sex (Men/Women) *	
Men	36 (62)
Women	22 (38)
Tumor size (cm) *	3.3 ± 1.4
Tumor location †	
Head	33 (57)
Neck	7 (12)
Body	12 (21)
Tail	6 (10)
Surgery method † (*n* = 19)	
Whipple operation	14
Distal pancreatectomy	2
Exploratory laparotomy and biopsy	1
Bypass and biopsy	2
TNM staging †	
I	4 (7)
II	12 (21)
III	9 (16)
IV	33 (57)
Histology grades † (*n* = 20)	
Well-differentiated	3
Moderately differentiated	8
Poorly differentiated	9

Note: * Data are the means ± standard deviations or median values, with range in parentheses. † Data are numbers of patients, with percentages in parentheses.

**Table 2 diagnostics-11-01541-t002:** Comparison of MRS metabolites between normal pancreas and pancreatic cancer.

Metabolites	ppm	Normal Pancreas	Pancreatic Cancer	*p* Value
Cr_2	3.9	11.2 ± 19.9	7.7 ± 13.9	0.200
Cr_1	3.03	8.9 ± 15.8	4.5 ± 6.1	0.076
Glx	2.05–2.5	137.9 ± 201.4	66 ± 72.3	0.005 *
NAA	2.02	78.9 ± 100.6	24.1 ± 41.7	<0.001 *
Lipid_2	1.3	3119.4 ± 7427.6	1015.2 ± 1190.7	<0.001 *
Lipid_1	0.9	886.3 ± 1218.7	181.4 ± 240.2	<0.001 *

Note: Cr = Creatine, Glx = Glutamine and Glutamate, NAA = N-acetyl-aspartate, Lipid_2 = methylene (–CH_2_), and Lipid_1 = methyl (CH_3_). * indicates *p* value is significant < 0.05.

**Table 3 diagnostics-11-01541-t003:** Differences in MRS metabolites relative to pathological grades and TNM stage.

Metabolites	Pathological Grades			T Stage			N Stage			M Stage		
	Well- and Moderately (*n* = 11)	Poorly Differentiated (*n* = 9)	*p* Value	T1–3 (*n* = 26)	T4 (*n* = 32)	*p* Value	N0 (*n* = 14)	N1 (*n* = 44)	*p* Value	M0 (*n* = 25)	M1 (*n* = 33)	*p* Value
Cr_2	6 ± 5.7	6.3 ± 12.8	0.152	6.6 ± 9.2	8.2 ± 16.7	0.506	14.2 ± 22.4	5.3 ± 8.9	0.019 *	5.6 ± 6.7	8.9 ± 17.3	0.85
Cr_1	5.3 ± 6.5	2.6 ± 3.4	0.370	4.7 ± 6.1	4 ± 6.1	0.681	4.7 ± 6.2	4.2 ± 6.1	0.581	4.4 ± 5.5	4.3 ± 6.6	0.395
Glx	91.4 ± 87.5	46.6 ± 39.6	0.230	76.4 ± 75	57.5 ± 70.1	0.152	85.8 ± 83.4	59.7 ± 68.3	0.170	69.2 ± 70.5	63.6 ± 74.7	0.392
NAA	22.0 ± 21.9	15.1 ± 15.7	0.603	32.1 ± 55.8	17.1 ± 19.6	0.359	27.4 ± 28	22.7 ± 43.8	0.271	16.9 ± 15.6	29 ± 51.5	0.519
Lipid_2	1337.6 ± 1278.2	1044.9 ± 1205.1	0.456	1139.3 ± 1246.1	860.4 ± 1128.6	0.101	1410.8 ± 1444	850.1 ± 1067.2	0.122	1088.9 ± 1192.3	907 ± 1183.6	0.282
Lipid_1	197.2 ± 274.9	289.6 ± 264.8	0.503	222.3 ± 274.4	138.4 ± 200	0.038 *	248.4 ± 253.7	153 ± 230.7	0.084	192.6 ± 213.6	163.5 ± 257	0.094

Note: Cr = Creatine, Glx = Glutamine and Glutamate, NAA = N-acetyl-aspartate, Lipid_2 = methylene (–CH_2_), and Lipid_1 = methyl (CH_3_). * indicates *p* value is significant < 0.05.

**Table 4 diagnostics-11-01541-t004:** Univariable analysis of prognostic factors for PFS and OS.

		PFS						OS					
		Univariable			Multivariable			Univariable			Multivariable		
Parameters	Cutoff	HR	95% CI	*p* Value	HR	95% CI	*p* Value	HR	95% CI	*p* Value	HR	95% CI	*p* Value
Age (y/o)	64.5	1.001	0.574–1.748	0.996				1.025	0.588–1.787	0.935			
Sex (Women vs. men)		1.222	0.691–2.162	0.490				0.731	0.411–1.301	0.287			
Size (cm)	3.1	0.712	0.408–1.243	0.232				0.965	0.554–1.683	0.821			
TNM stage (4 vs. ≦ 3)		2.096	1.164–3.774	0.014 *	2.084	1.11–3.92	0.023	1.858	1.038–3.326	0.037 *			
MRS metabolites													
Cr_2	10.26/12.02	0.32	0.115–0.895	0.022 *	0.299	0.10–0.86	0.025	0.309	0.096–1.001	0.039 *			
Cr_1	1.89/1.89	0.466	0.256–0.846	0.01 *				0.313	0.17–0.576	<0.0001 *	0.342	0.185–0.632	<0.0001
Glx	45.36/39.86	0.472	0.261–0.854	0.011 *				0.449	0.254–0.794	0.0048 *			
NAA	30.62/30.62	0.519	0.22–1.222	0.13				0.437	0.185–1.029	0.052			
Lipid_2	1720/835.7	0.496	0.22–1.117	0.084				0.489	0.27–0.887	0.016 *			
Lipid_1	149.25/133.35	0.452	0.24–0.851	0.012 *	0.575	0.295–1.117	0.102	0.478	0.261–0.873	0.014 *	0.556	0.302–1.023	0.059

Note: Cr = Creatine, Glx = Glutamine and Glutamate, NAA = N-acetyl-aspartate, Lipid_2 = methylene (–CH_2_), Lipid_1 = methyl (CH_3_), HR = hazard ratio, and CI = confidence interval. * indicates *p* value is significant < 0.05.

## Data Availability

The datasets generated during and/or analysed during the current study are available from the corresponding author upon reasonable request.

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
