# Peer review of "Metabolic Alterations in Pancreatic Cancer Detected by In Vivo 1H-MR Spectroscopy: Correlation with Normal Pancreas, PET Metabolic Activity, Clinical Stages, and Survival Outcome"

_diagnostics, 2021, doi:10.3390/diagnostics11091541_

Round 1

Reviewer 1 Report

In vivo magnetic resonance spectroscopy (MRS) is a difficult radiological assay due to inconsistencies that can occur between measurements due to low signal to noise and large linewidths.   The linewidth directly affects both the resolution and signal to noise of each spectral resonance. More information on the differences between linewidths between measurements is needed.   Signal to noise could possibly be much worse in the pancreatic cancer voxels because of heterogeneity in the tissue, which leads to inhomogeneity of the magnetic field.   Figure 2 lipid peak illustrates well the difference in linewidth between the two voxels. The authors could address this issue by providing the mean with the standard deviation of the overall linewidth for the water peak observed after shimming in the pancreatic cancer voxels versus normal pancreas.   In addition, if the authors did a test/retest analysis on some of the patients illustrating the precision of the analysis that would help relieve some of the doubt in the measurement. Also, was the same analysis performed on a phantom to illustrate the accuracy of the measurement.   The paper would be stronger if the same analysis performed in Table 2, Table 3, and Table 4 was performed again with the PET parameters measured in patients. This would allow the reader to compare directly the statistical analysis of clinical outcomes to the MRS observations and to the PET observations.   The paper would also be stronger if they could correlate metabolite concentration with IHC staining of proteins associated with a particular metabolite, such as N-acetylaspartate with NAT8L.   However not required, the paper would be stronger if they have ex vivo magnetic resonance spectroscopy of biopsy or resected pancreatic tissue that illustrates the same reduction in creatine, NAA or lipids as observed by in vivo MRS.  

Other small issues – The text in lines 120 to 125 does not need to be in italics.  In line 327, I would change the text to “recent research.”

Reviewer 2 Report

The autors present a retrospective study of metabolic imaging using NMR-spectroscopy and FDG-PET in pancreatic cancer. In short, some metabolites are demonstrated to correlate with tumor metabolic activity and survival. Creatinine even qualifies as prognostic in multivariable analysis adjusted for clinical staging, age, sex etc.

Introduction, methods, presentation of results and interpretation are appropriate.

A special merit of the study is that it identifies metabolic markers that could be measured non-invasively to predict and monitor treatment response in conventional chemotherapy as well as novel metabolic targeted therapies.

In summary, I recommend publication.

Minor point: As a surgeon, I would like to know if there were metabolic differences in patients with R0 and R1/R2 resection ?

Author Response

Dear editor and reviewer:

We deeply appreciate the positive comments from the reviewer.

Point 1: As a surgeon, I would like to know if there were metabolic differences in patients with R0 and R1/R2 resection?

Response 1: In patients receiving curative surgery (n =16), 7 patients had R0 resection and 9 patients had R1/2 resection.

Lower lipid_1 level was found in patients with R1/2 resection (vs. R0, p = 0.023). There were no statistical differences of other metabolites between R0 and R1/2 resection.

We thank the reviewer for providing this study idea and agree this finding may be useful for surgeons. We did not present this result in the manuscript because the patient number is too small. In the future, we will include more patients with curative surgery to confirm this result.

Round 2

Reviewer 1 Report

Previous Review Point 6: The paper would be stronger if the same analysis performed in Table 2, Table 3, and Table 4 was performed again with the PET parameters measured in patients. This would allow the reader to compare directly the statistical analysis of clinical outcomes to the MRS observations and to the PET observations.  

The response to this point in the cover letter does not answer the question – the PET study was performed on all the same patients as MRS based on the methods section. How does SUVmax, MTV, and TLG correspond with well and moderately differentiation versus poorly differentiation; T1-3 versus T4; N0 versus N1; and M0 versus M1.   How does SUVmax, MTV and TLG correlate to progression free survival and overall survival.

Methods section: Could you clarify more on your voxel sizes and provide more detail.   Was the size of the voxel used for every specific patient for pancreatic cancer versus normal pancreas the same (line 111) but, the size of the voxel used for each patient different (line 125, range of voxel sizes)?   Did you run any analysis to illustrate that the metabolic values observed were not correlated to voxel size?      

Line 111 in paper:   Single-voxel MRS data were acquired by replacing the volume of interest in pancreatic cancer and normal pancreatic parenchyma, with careful avoidance of inclusion of adjacent vessels.

Line 125 in the paper: The volumes of interest were 3.6 ± 1.3 cm3 (range, 3.4–12.5 cm3) for tumor and 3.9 ± 1 cm3 (range, 1.9–7.7 cm3) for normal pancreatic parenchyma.

Discussion: In your section about the complex nature of in vivo MRS and its inconsistencies – I would add how air, water, and tissue interfaces specifically in the torso make it difficult to generate a homogenous magnetic field. Include a paragraph on the difficulties in standardizing the methodology across institutions and how you could possibly overcome them. Please elaborate more on how you think the MRS information could be used in the clinic.

Small points:

Remove bold text in lines 130 and 131.   Remove much in line 331.   
